# Hotspots and Season Related to Wildlife Roadkill in the Amazonia–Cerrado Transition

**Evandro Santos [1], Milton Cordova [2] , Clarissa Rosa [3,4,* and Domingos Rodrigues [1,4]**

[1] Postgraduate Program in Environmental Sciences, Institute of Natural, Human and Social Sciences, Federal University of Mato Grosso, Avenue Alexandre Ferronato, 1200, Sinop 78550728, Brazil

[2] Postgraduate Program in Botany, Department of Botany, Institute of Biological Sciences, University of Brasília, Brasília 70910900, Brazil

[3] Biodiversity Coordination, National Institute for Amazon Research, Manaus 69067375, Brazil

[4] National Institute of Science and Technology for Integrated Studies of Amazon Biodiversity, Ministry of Science and Technology, National Council for Scientific and Technological Development, Sinop 78555901, Brazil

* Correspondence: rosacla.eco@gmail.com

**Abstract:** The construction of new roads opens access to native environments, resulting in changes to the landscape. These roads cause the death of native wildlife due to collisions with vehicles, which is the main cause of human-induced vertebrate death. This work aimed to investigate the spatial distribution of roadkills on the BR-163 highway, Mato Grosso, Brazil, an Amazonia–Cerrado transition zone, to identify roadkill hotspots. The study area consisted of 244 km of road. Twenty-five trips were made totaling 6100 km of surveyed roadway between 2008 and 2011. A total of 1005 individuals from 65 species was recorded in this study. Mammals were the most affected, both in number of individuals and species, followed by birds, reptiles, and amphibians. The species with the highest roadkill rates were *Cerdocyon thous* and *Tyto furcata*. Identified endangered mammals included *Myrmecophaga tridactyla*, *Chrysocyon brachyurus*, *Priodontes maximus*, *Speothos venaticus*, *Tapirus terrestris*, *Pteronura brasiliensis*, and *Ateles marginatus*. The highest rates of roadkill were recorded during the wet season. The location and number of hotspots varied among classes and were related to different environmental variables such as waterbodies, forest fragments, and agricultural areas. Roadkills may be a major threat for vulnerable species and should therefore be studied to define conservation plans for local species and road traffic.

**Keywords:** hotspots; road ecology; animal behavior; mammals; birds; reptiles; amphibians

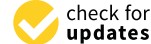



## 1. Introduction

Land-use change due to agriculture and livestock expansion, together with infrastructure implementation, has caused severe biodiversity loss in the Amazonia–Cerrado transition zone [1,2]. The state of Mato Grosso has become one of the most active deforestation frontiers in Amazonia in recent decades [3,4]. Most of this deforestation has been caused by cattle farming and increased soy production in the early 2000s [5,6]. In 2016, agribusiness contributed more than 45% of the domestic product of Mato Grosso, which is the leading state of Brazil (GDP) (IMEA 2018), producing 27.25% of the country's total grain within an area of 9.52 million hectares according to CONAB (https://www.conab.gov.br/info-agro/safras/graos/ (accessed on 1 February 2022)). Transportation of the grain is carried out on roads, mostly to the port of Santos, SP (approximately 45%), and nearby ports in the State of Pará (30%) [7]. The BR-163 highway (Cuiabá, MT-Santarém, PA) was constructed in the early 1970s [8] and is still the main access route through the central and northern regions of Mato Grosso.

The construction of roads opens access to natural environments and is directly linked to significant man-made changes in landscape [9,10], where approximately 85% of deforestation occurs within fifty kilometers of the road boundary [11]. There are currently

more than 1.72 million kilometers of highway in Brazil according to the National Transport Confederation (CNT) (http://anuariodotransporte.cnt.org.br/2018/Rodoviario/1-1-/Principais-dados (accessed on 18 June 2022)), which may have directly caused the destruction of more than 595,000 hectares of vegetation previously occupied by a diverse range of organisms [12]. It was estimated that 25 million kilometers of new roads will be built worldwide by 2050, an increase of 60% in the total length of roads compared to 2010, with 90% of them to be built in developing countries such as Brazil [13]. The impacts on wildlife are manifold, including changes in animal behavior and habitat use [14–16]; modified composition, structure, and dynamics of communities [17,18]; and increased mortality rate [19,20].

Wildlife roadkill is one of the most visible and studied impacts of roads and is considered the main cause of vertebrate death by human activity [21]. In Brazil, it is estimated that 473 million vertebrates are killed by collisions with vehicles each year [22]. Generally, roads with intense traffic located in areas of high biodiversity tend to have high rates of roadkill [23–25]. Roadkill hotspots and deforestation can worsen the predicament of endangered species such as the giant anteater (*Myrmecophaga tridactyla* Linnaeus, 1758), the maned wolf (*Chrysocyon brachyurus* (Illiger, 1815), and the jaguar (*Panthera onca* (Linnaeus, 1758)), as the population densities of these species are already low and are further reduced due to deaths resulting from vehicle collisions, which ultimately results in loss of biodiversity [26]. In addition, the barrier effect caused by roads can isolate populations and prevent access to resources such as water, food, and shelter [27,28] and cause gene flow loss in populations [29].

Environmental seasonality affects the frequency of roadkills [30]. Many studies point out that they may occur more frequently in the wet season [31–34] than in the dry season [23,35,36]. The numbers of reptile and amphibian roadkills are commonly higher in the wet season as this is the period of greatest activity for these groups [37]. Meanwhile, birds require a greater sampling effort than the other groups [38] and therefore rarely present any pattern related to seasons [30]. Medium and large mammals are generally not influenced by the seasons [39], except for carnivores, which show seasonal patterns related to specific phases of their life cycles (e.g., birth, dispersal, and reproduction) [24].

To lessen the impacts on fauna, planning and implementation of mitigation measures are necessary [40]. Tunnels and bridges for faunal crossings, road traffic devices to slow vehicles, signage, and other actions have been used more frequently in recent decades [41]; however, their efficiency depends on the location where they are used [40]. Fauna crossings are considered the main measures, as they enable gene flow between fragmented forest areas [24,41]. Monitoring roadkill can positively contribute to mitigation recommendations, including the installation of fauna passages [39].

In addition, monitoring roadkill can serve as an additional indicator of local diversity providing ecological information such as species composition and natural history [39,42]. This information makes it possible to assess the degree of conservation of a site and establish priority areas for habitats and species conservation [13,43]. Our work therefore aimed to investigate the spatial distribution of roadkill on the BR-163 highway between Lucas do Rio Verde and Itaúba, Mato Grosso, and to identify roadkill hotspots by answering the following questions: (1) Is there any variation in the rate of roadkill between the wet and dry seasons? (2) Does the landscape have any effect on animal deaths? (3) Are there endangered species within the study area that have suffered road mortality? (4) Does the location of roadkill hotspots vary between the different groups sampled?

## 2. Materials and Methods

### 2.1. Study Area

The study was carried out on the BR-163 highway (latitude 13°03′49,46′′ S–longitude 55°55′11,56′′ W and latitude 11°00′23,77′′ S–longitude 55°05′02,32′′ W) between Lucas do Rio Verde (km 689) and Itaúba (km 933), northern Mato Grosso (244 km of road monitored). The region's climate is described under the Köppen climate classification system as Am: hot

and humid, a transition between the superhumid equatorial climate (Af) of Amazonia and the humid tropical climate (Aw) of the Central Plateau [44]. According to Souza et al. [45], 94.51% of the annual rainfall occurs between the months of October and April (wet season), and 5.49% falls between May and September (dry season). The annual average rainfall and temperature are 1974 mm and 24.7 °C, respectively [45].

Data on roadkill and species were collected on BR-163, a dual-lane single carriageway with an asphalt surface and no side structures that is predominantly straight with a few winding stretches (Figure 1). The estimated average daily number of vehicles between Sorriso and Sinop ranged from 3929 to 4757 between 2008 and 2011 (data obtained from Federal Highway Police), with more than 50% of these vehicles being cargo transport [46].

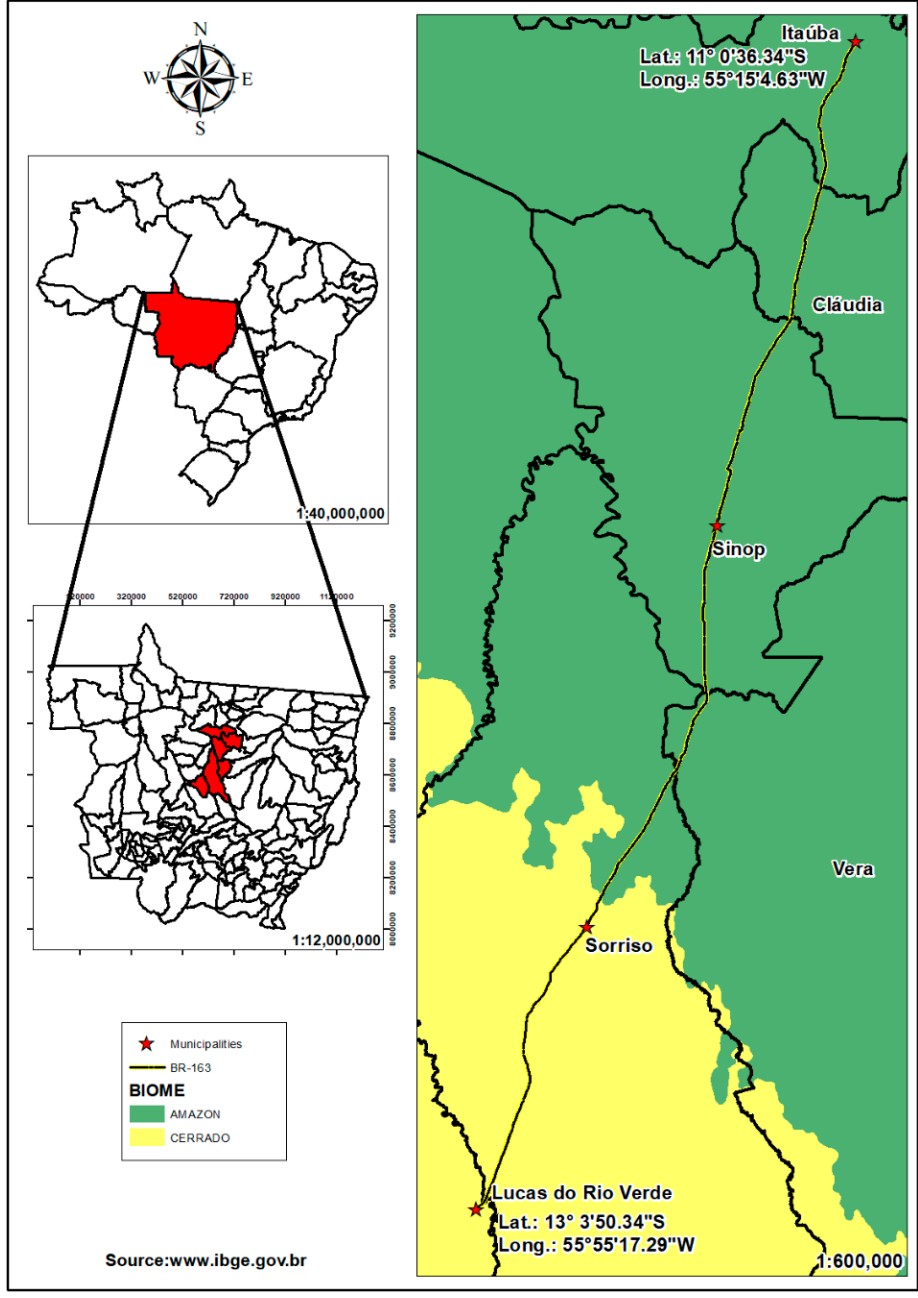

**Figure 1.** Location of the monitored stretch of road BR-163, between the municipalities of Lucas do Rio Verde and Itaúba, Mato Grosso, Amazonia–Cerrado transition zone, Brazil.

The study area is located within a transition zone between Amazonia and Cerrado. The Brazilian Amazonia contains 30% of the world's tropical forest [47] and the highest biological diversity, housing approximately 10% of plant species and between 6% and 10% of vertebrate species [48]. The Cerrado biome has the richest biodiversity among existing savannas and is considered an important "hotspot" for biodiversity conservation [49,50]. In Mato Grosso, the transition from the Brazilian Amazonia to Cerrado creates an overlap of both plant assemblages, resulting in high local biodiversity [51,52]. This high diversity of both fauna and flora suffers from the economic and demographic growth of the state [53], especially in Amazonia and the Amazonia–Cerrado transition zone [54–56].

### 2.2. Data Collection

Data on roadkills were recorded between September 2008 and June 2011. During this period, 25 trips were carried out to monitor wildlife road mortality (2 trips in 2008, 11 in 2009, 8 in 2010, and 4 in 2011), totaling 6100 km of monitored road. Collections were carried out by three observers to improve the accuracy of the survey, keeping the average speed of the vehicle at 40 km/h. Monitoring started at 06:00 with the objective of spotting as many animal carcasses as possible that had been road-killed the night before, thus preventing them from being destroyed by vehicle traffic or scavengers. Sampling ended at approximately 16:00. The monitoring trips were carried out in alternate directions, with animals being counted only on the way.

The geographical position of each roadkill was recorded using a GPS device. A photographic record was also made for later identification to a specific level using the literature [57–60] and consultation with specialists. The carcasses of small and medium-sized animals were removed to reduce the risk of other animals, such as scavengers, being road-killed. Information was collected for each record regarding the local landscape (area, m$^2$) of pasture/crops, water bodies/wetlands, or native vegetation to enable analysis of roadkill distribution patterns. In addition, satellite images were used to characterize the type of environment for each roadkill occurrence. Satellite images were analyzed in the QGIS program (Quantum GIS—Geographic Information System—Version 3.4.7, https://download.qgis.org/downloads (accessed on 18 May 2022)) and cross-checked with information collected in the field.

To assess the occurrence of road-killed endangered species, the Official List of Endangered Brazilian Fauna Species (https://in.gov.br/en/web/dou/-/portaria-mma-n-148-de-7-de-junho-de-2022-406272733 (accessed on 19 June 2022)) and the IUCN Red List of Threatened Species (https://www.iucnredlist.org/ (accessed on 18 June 2022)) were used. Collection of the carcasses (freshly killed animals) was permitted by the SISBIO license number 12636.

### 2.3. Data Analysis

To assess the difference between the total number of roadkills by class (Mammalia, Aves, Reptilia, and Amphibia) and the wet and dry seasons, a generalized linear model (GLM) test with negative binomial distribution was performed using R software (ver. 3.2.6) [61] with the function glm.nb in the MASS package [62]. The adopted distribution was selected by the Akaike Information Criteria (AIC) [63].

To allow comparison with other works, the rates of roadkill were calculated as suggested by Bager and Rosa [38] using Siriema v2.0 software [64].

To identify the landscape variables that best explain roadkills, the GLM test with negative binomial distribution was carried out using R software [61].

To identify roadkill hotspots and their spatial distribution, the Siriema v2.0 program 2D Ripley K statistics test was used [64]. A one-kilometer radius was used in the 2D hotspot identification test (2D Ripley K statistics test: 95% confidence limit and one thousand simulations), as this radius value is suitable for the implementation of mitigation measures aiming to increase biodiversity [40].

The Siriema program allows to maintain the two-dimensionality of the route (Ripley's K statistic—2D) as proposed by Coelho et al. [39], which reduces the error, as road lineariza- tion can consider greater distances for records that could be spatially close on winding highways [64]. According to Coelho et al. [64], the 2D Ripley's K test uses a radius centered on a roadkill record forming a circle, adding up the other records within this area. This analysis can be described by the equation below:

$$K(r) = \frac{D}{n(n-1)} \sum_{i=1}^{n} 2r/Ci(r) \sum_{j \neq i} fij$$

where $D$ = highway length; $n$ = number of records; $r$ = radius; $i$ = event; $j$ = other event; $Ci(r)$ = length of road inside the circle of radius $r$ centered on event $i$; and $fij$ = 0 if $j$ is outside the circle of radius $r$ centered on $i$, or 1 if $j$ is inside that area.

The non-randomness of the spatial distribution for events at different scales can be assessed using Ripley's K statistic [65,66]. To interpret this test, the L(r) function is used, which allows for analyzing grouping intensity at different scales [65,66], as proposed by Clevenger et al. [67] and modified by Coelho et al. [39].

$$L(r) = K(r) - Ks(r)$$

where L(r) = difference between the observed K statistic value for the r scale and a simulated average K value for the r scale and Ks(r) = mean K values in simulations of randomly distributed roadkill records.

L(r) values above the confidence limits show scales with significant groupings, and values below these limits show scales with significant dispersion [64]. The initial radius used was 100 m, with a radius increment of 500 m (488 segments), one thousand simulations, and a confidence limit of 95% (modified from Coelho et al. [39], Cáceres et al. [33], and Teixeira et al. [40]).

These analyses were carried out for the classes separately and for all classes together. The 2D hotspot identification test was used to identify the places with the highest occurrence of roadkills (hotspots). The N events—N simulated function allows to point out significant grouping [65]. Unidentified carcasses were not used in the statistical analyses.

## 3. Results

Throughout the study (2008–2011), 1005 vertebrate roadkills were recorded, belonging to 65 species, comprising 355 (35.32%) mammals, 319 (31.74%) birds, 257 (25.57%) reptiles, 51 (5.07%) amphibians, and 23 (2.29%) that were not identified due to the level of decompo- sition of the carcasses (Table 1). A total of 83.10% of the mammals were identified to the species level: 78.99% of birds, 44.74% of reptiles, and 96.08% of amphibians, which had two dominant species, *Leptodactylus labyrinthicus* (62.74%) and *Rhinella marina* (33.33%).

On the monitored road, 0.16 animals/km/day and 1.98 animals/km/year suffered road mortality. The roadkill rate per trip was highest for mammals (0.06 individuals/km/day), followed by birds (0.05 individuals/km/day), reptiles (0.04 individuals/km/day), and amphibians (0.01 individuals/km/day).

For mammals, crab-eating fox (*Cerdocyon thous*, 95 individuals), six-banded armadillo (*Euphractus sexcinctus*, 58 individuals), capybara (*Hydrochoerus hydrochaeris*, 38 individuals), and collared anteater (*Tamandua tetradactyla*, 31 individuals) totaled 222 individual roadkills, corresponding to 62.53% and 22.09% of mammals and total fauna killed by vehicle collision, respectively. Seven endangered species were killed: giant anteater (*Myrmecophaga tridactyla*), maned wolf (*Chrysocyon brachyurus*), giant armadillo (*Priodontes maximus*), South American tapir (*Tapirus terrestres*), bush dog (*Speothos venaticus*), giant otter (*Pteronura brasiliensis*), and white-cheeked spider monkey (*Ateles marginatus*) (Table 1).

**Table 1.** Roadkill and conservation status of Brazilian fauna along the 244 km long stretch of the BR-163 highway, Mato Grosso, Brazil. The conservation status of the species follows the Official List of Endangered Brazilian Fauna Species and IUCN. * Endangered species. N.I.—unidentified.

| Class | Family | Scientific Name | Number of Individuals | Roadkill Rate Ind./km/day | Status |
|---|---|---|---|---|---|
| Amphibia | Bufonidae | *Rhinella marina* (Linnaeus, 1758) | 17 | 0.0028 | LC |
| | Leptodactylidae | *Leptodactylus labyrinthicus* (Spix, 1824) | 32 | 0.0052 | LC |
| | Ampibian n.i | Amphibian N.I. | 2 | 0.0003 | - |
| Aves | Anatidae | *Anas* sp. | 1 | 0.0002 | - |
| | Ardeidae | *Butorides striata* (Linnaeus, 1758) | 1 | 0.0002 | LC |
| | Cathartidae | *Coragyps atratus* (Bechstein, 1793) | 2 | 0.0003 | LC |
| | Charadriidae | *Vanellus chilensis* (Molina, 1782) | 1 | 0.0002 | LC |
| | Columbidae | *Columbina talpacoti* (Temminck, 1810) | 10 | 0.0016 | LC |
| | | *Patagioenas picazuro* (Temminck, 1813) | 3 | 0.0005 | LC |
| | Cuculidae | *Crotophaga ani* (Linnaeus, 1758) | 29 | 0.0048 | LC |
| | | *Guira guira* (Gmelin, 1788) | 8 | 0.0013 | LC |
| | Falconidae | *Caracara plancus* (Miller, 1777) | 3 | 0.0005 | LC |
| | Passeridae | *Passer domesticus* (Linnaeus, 1758) | 5 | 0.0008 | LC |
| | Psittacidae | *Eupsittula aurea* (Gmelin, 1788) | 6 | 0.0010 | LC |
| | Ramphastidae | *Pteroglossus viridis*(Linnaeus, 1766) | 1 | 0.0002 | LC |
| | Rheidae | *Rhea americana* (Linnaeus, 1758) | 5 | 0.0008 | NT |
| | Strigidae | *Athene cunicularia* (Molina, 1782) | 36 | 0.0059 | LC |
| | Thraupidae | *Tangara cayana* (Linnaeus, 1766) | 2 | 0.0003 | LC |
| | | *Volatinia jacarina* (Linnaeus, 1766) | 13 | 0.0021 | LC |
| | Tinamidae | *Crypturellus* sp. | 3 | 0.0005 | - |
| | | *Rhynchotus rufescens* (Temminck, 1815) | 42 | 0.0069 | LC |
| | Tyrannidae | *Pitangus sulphuratus* (Linnaeus, 1766) | 2 | 0.0003 | LC |
| | Tytonidae | *Tyto furcata* (Temminck,1827) | 83 | 0.0136 | LC |
| | Bird N.I. | Bird N.I. | 63 | 0.0103 | - |
| Mammalia | Atelidae | *Ateles marginatus* (É. Geoffroy, 1809) * | 1 | 0.0002 | EN |
| | Canidae | *Cerdocyon thous* (Linnaeus, 1766) | 95 | 0.0156 | LC |
| | | *Chrysocyon brachyurus* (Illiger, 1815) * | 2 | 0.0003 | VU |
| | | *Speothos venaticus* (Lund, 1842) * | 1 | 0.0002 | VU |
| | Caviidae | *Cavia aperea* Erxleben, 1777 | 2 | 0.0003 | LC |
| | | *Hydrochoerus hydrochaeris* (Linnaeus, 1766) | 38 | 0.0062 | LC |
| | Cebidae | *Sapajus apella* (Linnaeus, 1758) | 4 | 0.0007 | LC |
| | Chiroptera | Bats N.I. | 9 | 0.0015 | - |
| | Cricetidae | *Bolomys* sp. | 3 | 0.0005 | - |
| | Cuniculidae | *Cuniculus paca* (Linnaeus, 1766) | 2 | 0.0003 | LC |
| | Dasypodidae | *Priodontes maximus* (Kerr, 1792) * | 2 | 0.0003 | VU |
| | | *Dasypus novemcinctus* Linnaeus, 1758 | 28 | 0.0046 | LC |
| | | *Euphractus sexcinctus* (Linnaeus, 1758) | 58 | 0.0095 | LC |
| | Dasypodidae | Armadillo N.I. | 12 | 0.0020 | - |
| | Didelphidae | *Didelphis* sp. | 14 | 0.0023 | - |
| | Erethizontidae | *Coendou prehensilis* (Linnaeus, 1758) | 15 | 0.0025 | LC |
| | Felidae | *Leopardus pardalis* (Linnaeus, 1758) | 2 | 0.0003 | LC |
| | | *Puma concolor* (Linnaeus, 1771) | 1 | 0.0002 | LC |
| | Mustelidae | *Eira barbara* (Linnaeus, 1758) | 1 | 0.0002 | LC |
| | | *Lontra longicaudis* (Olfers, 1818) | 1 | 0.0002 | NT |
| | | *Pteronura brasiliensis* (Zimmermann, 1780) * | 1 | 0.0002 | EN |
| | Myrmecophagidae | *Myrmecophaga tridactyla* Linnaeus, 1758 * | 6 | 0.0010 | VU |
| | | *Tamandua tetradactyla* (Linnaeus, 1758) | 32 | 0.0052 | LC |
| | Procyonidae | *Nasua nasua* (Linnaeus, 1766) | 2 | 0.0003 | LC |
| | | *Procyon cancrivorus* (G. Cuvier, 1798) | 2 | 0.0003 | LC |
| | Tapiridae | *Tapirus terrestris* (Linnaeus, 1758)* | 2 | 0.0003 | VU |
| | Mammalia N.I. | Mammalia N.I. | 19 | 0.0031 | - |
| Reptilia | Alligatoridae | *Paleosuchus* sp. | 6 | 0.0010 | - |
| | Amphisbaenidae | *Amphisbaena alba* Linnaeus, 1758 | 4 | 0.0007 | LC |
| | Boidae | *Boa constrictor constrictor* (Linnaeus, 1758) | 58 | 0.0095 | LC |

**Table 1.** *Cont.*

| Class | Family | Scientific Name | Number of Individuals | Roadkill Rate Ind./km/day | Status |
|---|---|---|---|---|---|
| | | *Epicrates cenchria* (Linnaeus, 1758) | 12 | 0.0020 | LC |
| | | *Eunectes murinus* (Linnaeus, 1758) | 9 | 0.0015 | LC |
| | Colubridae | *Clelia plúmbea* (Wied, 1820) | 1 | 0.0002 | LC |
| | | *Oxyrrophus* sp. | 3 | 0.0005 | - |
| | | *Philodryas olfersii* (Lichtenstein, 1823) | 7 | 0.0011 | LC |
| | | *Spilotes pullatus* (Linnaeus, 1758) | 8 | 0.0013 | LC |
| | Elapidae | *Micrurus* sp. | 4 | 0.0007 | - |
| | Iguanidae | *Iguana iguana* (Linnaeus, 1758) | 1 | 0.0002 | LC |
| | Teiidae | *Ameiva ameiva* (Linnaeus, 1758) | 3 | 0.0005 | LC |
| | | *Tupinambis* sp. | 6 | 0.0010 | - |
| | Testudinidae | *Chelonoidis carbonarius* (Spix, 1824) | 2 | 0.0003 | NE |
| | Viperidae | *Bothrops atrox* (Linnaeus, 1758) | 2 | 0.0003 | LC |
| | | *Bothrops moojeni* Hoge, 1966 | 7 | 0.0011 | LC |
| | | *Crotalus durissus* Linnaeus, 1758 | 1 | 0.0002 | LC |
| | Reptilia N.I | Snake | 123 | 0.0202 | - |
| N.I. | N.I. | N.I. | 23 | 0.0038 | - |
| Total | | | 1005 | 0.1648 | |

(Endangered, EN; Vulnerable, VU; Near-Threatened, NT; Least Concern, LC; Not Applicable, NA; Not Evaluated, NE).

For birds, the species with the highest number of killed individuals were *Tyto furcata* (83 individuals), *Rhynchotus rufences* (42 individuals), and *Athene cunicularia* (36 individuals), which accounted for 50.47% of the total number of killed birds. Most reptiles killed during the study period were represented by individuals of the species *Boa constrictor constrictor* (58 individuals), *Epicrates cenchria* (12 individuals), and *Eunectes murinus* (nine individuals), totaling 79 individuals, representing 30.73% of all recorded reptiles. None of the birds, reptiles, or amphibians recorded are included in the endangered species list.

There were 764 recorded roadkills during the wet season and 241 during the dry season (Figure 2).

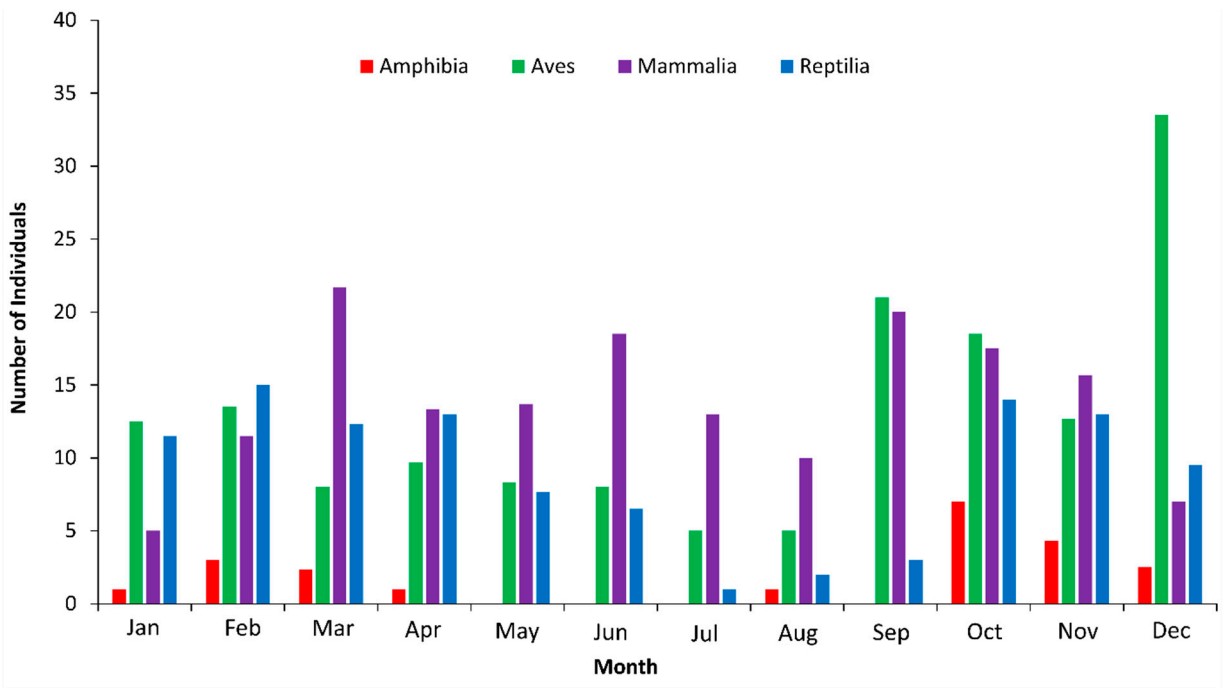

**Figure 2.** Monthly average number of roadkills by class on the BR-163 highway in the Amazonia–Cerrado transition zone, Mato Grosso, Brazil.

There was no significant difference in the number of roadkills between the wet and dry seasons for mammals. The number of roadkills was the highest in the wet season for amphibians (z = 2.068, *p* = 0.029), reptiles (z = 3.018, *p* = 0.002), and birds (z = 1.025, *p* = 0.041) (Figure 3).

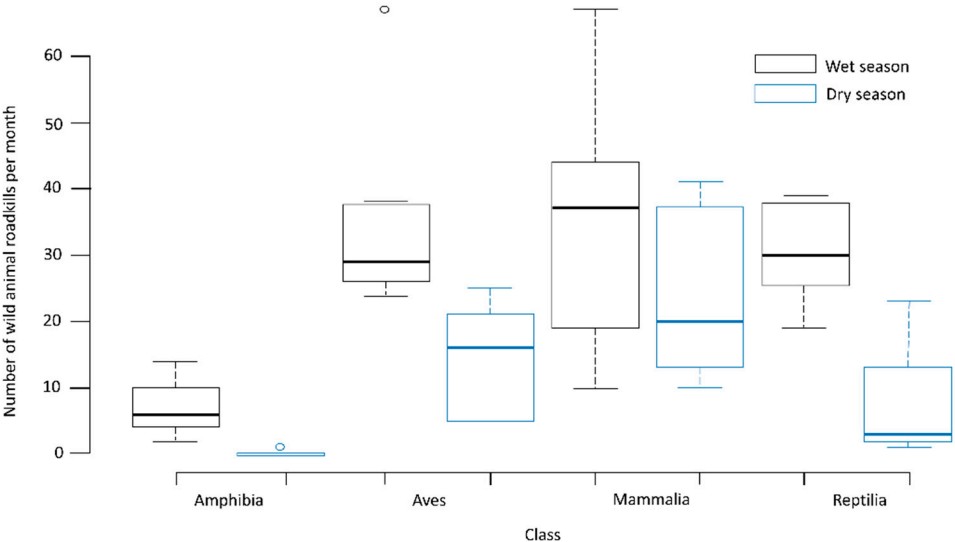

**Figure 3.** Average number of roadkills recorded by class in the wet and dry seasons on the BR-163 highway in the Amazonia–Cerrado transition zone, Mato Grosso, Brazil.

In general, our analyses showed that all classes combined have a positive relationship with forest fragments (z = 4.950, *p* < 0.0001; Figure 4) and a negative one with urban areas (z = −4.692, *p* < 0.0001; Figure 4). For Mammalia and Reptilia, we found a positive relationship with wetlands (z= 2.420 and *p* = 0.015 and z = 2.723 and *p* = 0.006, respectively; Figure 4). Birds showed a positive relationship with crops (z = 2.516, *p* = 0.0119; Figure 4).

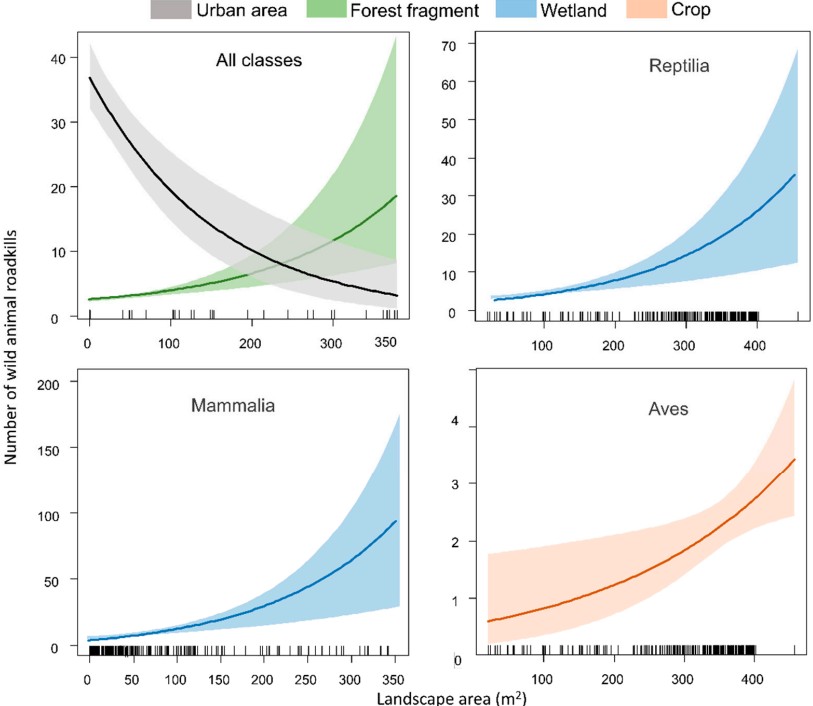

**Figure 4.** Environmental factors affecting roadkill rates of all classes combined and separately on the BR-163 highway in the Amazonia–Cerrado transition zone, Mato Grosso, Brazil.

Significant groupings of roadkills were found for all classes combined (Figure 5) and for the classes Mammalia, Aves, Reptilia, and Amphibia analyzed individually (Figure 6).

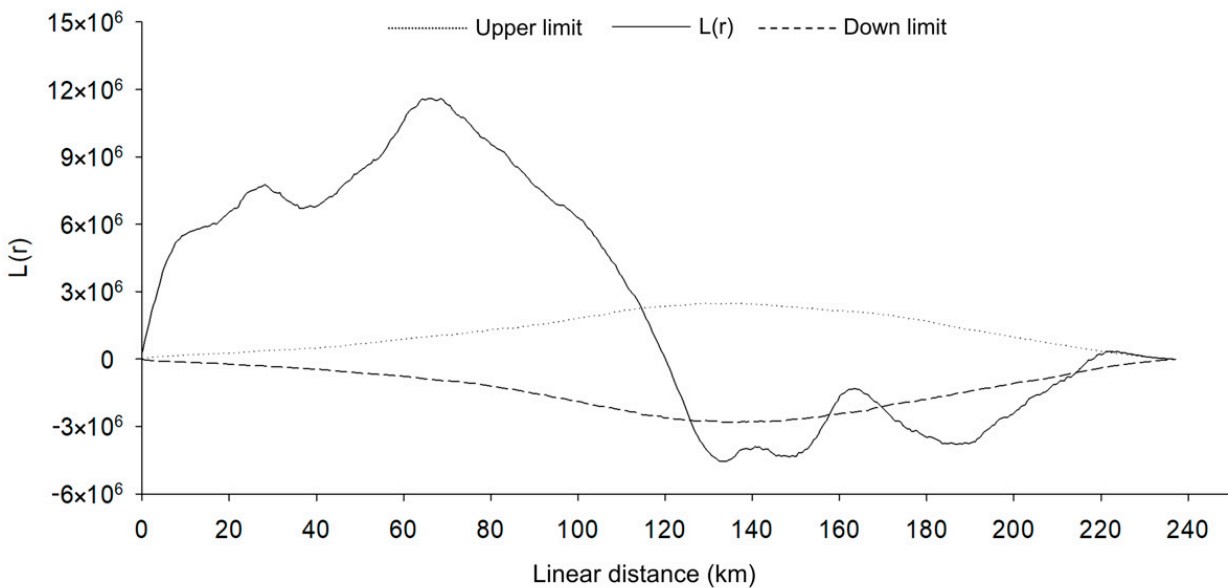

**Figure 5.** Roadkill groupings according to the 2D Ripley K statistics test for all animals on the BR-163 highway in the Amazonia–Cerrado transition zone, Mato Grosso, Brazil.

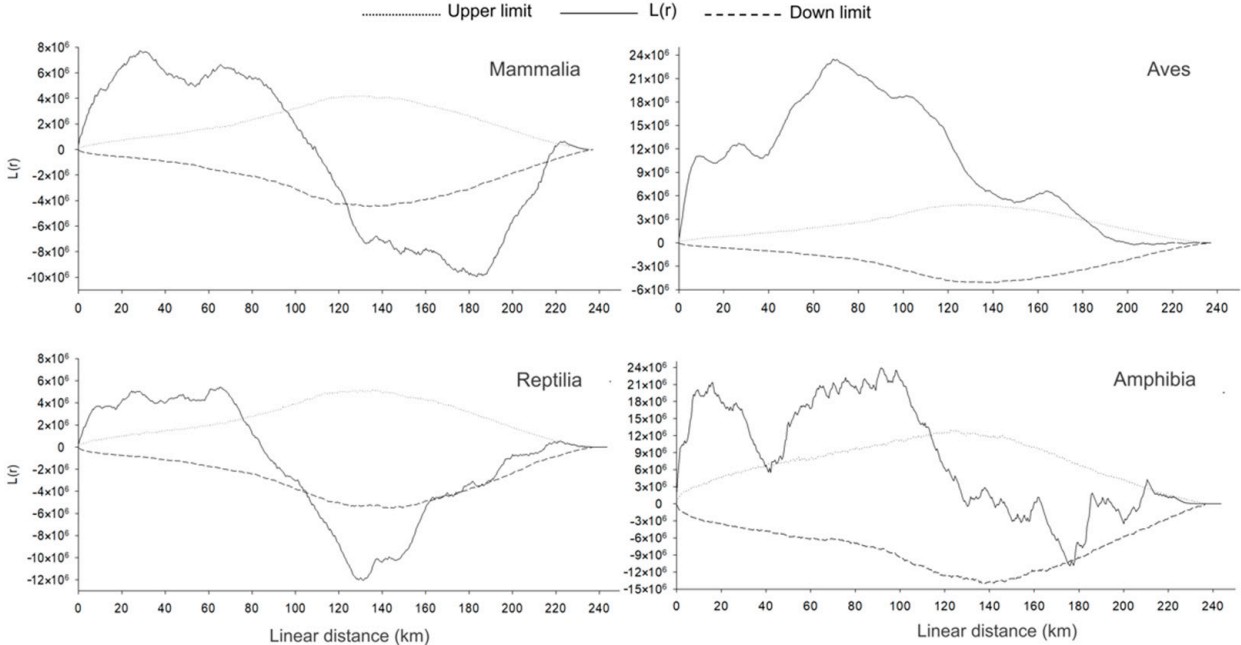

**Figure 6.** Roadkill groupings according to the 2D Ripley K statistics test for the classes Mammalia, Aves, Reptilia, and Amphibia on the BR-163 highway in the Amazonia–Cerrado transition zone between the municipalities of Lucas do Rio Verde and Itaúba, Mato Grosso, Brazil.

Most roadkill groupings occurred in the first 120 km, covering Lucas do Rio Verde, Sorriso, and part of Sinop (Figure 7). Peaks were observed between kilometers 10 and 12, 22 and 27, 32 and 35, 68 and 78, and 92 and 95. Another peak was recorded between 230 and 235 km near Teles Pires River, located in the Amazonia domain, Itaúba municipality.

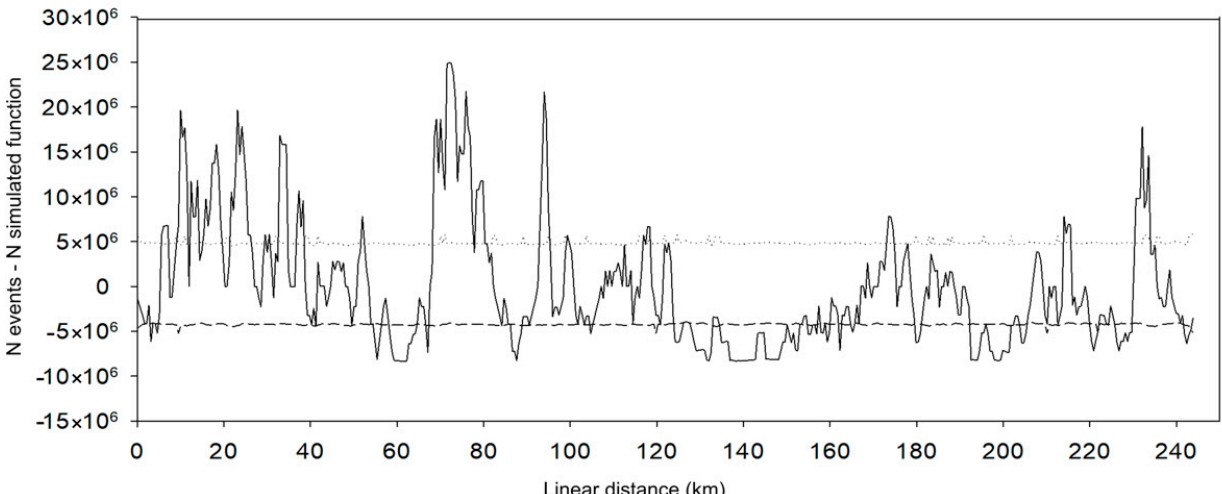

**Figure 7.** Location of roadkill hotspots on the BR-163 highway in the Amazonia–Cerrado transition zone, Mato Grosso, Brazil. Solid black line: N events—N simulated function. Dotted line: upper confidence limit; dashed line: lower confidence limit.

For mammals, hotspots occurred between kilometers 230 and 235 (Figure 8). The landscape surrounding roadkill records and roadkill hotspots was identified as forest fragments with nearby trails or watercourses.

For birds, most roadkill hotspots occurred in the first 100 km of the highway, with greater intensity between kilometers 70 and 80 (Figure 8). The landscape around these hotspots is characterized by the presence of crops or pasture, with small fragments of forest.

Roadkill hotspots for reptiles were distributed along the entire route, with peaks between kilometers 33 and 34, 70 and 73, 92 and 94, and 169 and 171 (Figure 8). The landscape around these hotspots is largely characterized by the presence of watercourses or wetlands associated with forest fragments.

Roadkill hotspots for amphibians occurred in a few sections of the studied route, where the greatest intensity occurred between kilometers 17 and 20 and 24 and 26 (Figure 8). Although the records showed a positive relationship with forest fragments only, the landscape around these hotspots was, for the most part, characterized by the presence of crops with forest fragments associated with watercourses.

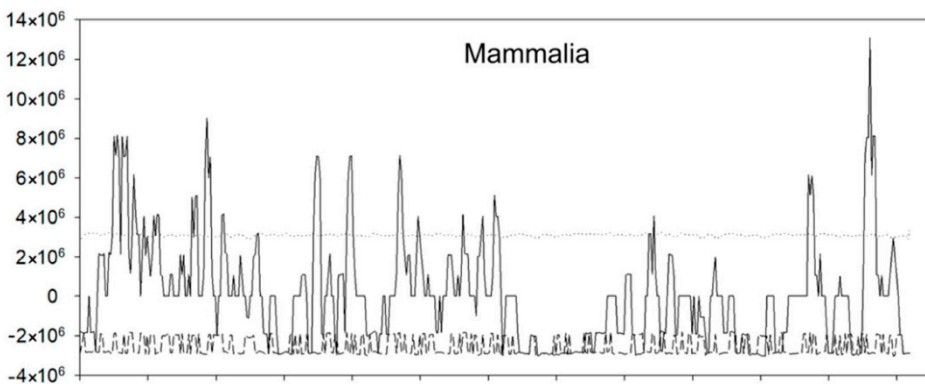

**Figure 8.** *Cont*.

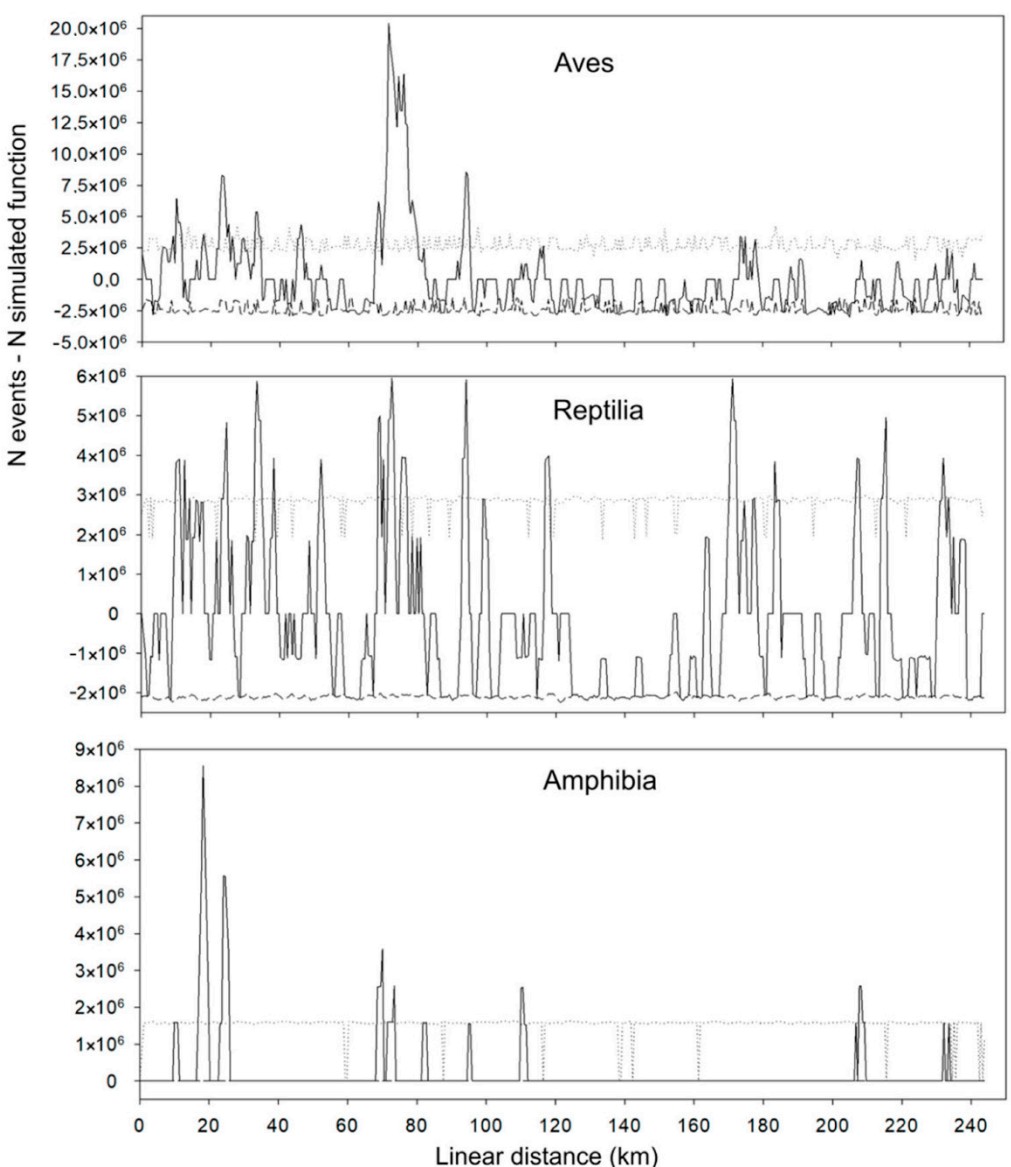

**Figure 8.** Location of roadkill hotspots for the classes Mammalia, Aves, Reptilia, and Amphibia on the BR-163 highway in the Amazonia–Cerrado transition zone, Mato Grosso, Brazil. Solid black line: N events—N simulated function; dotted lines: upper confidence limit; dashed lines: lower confidence limit.

## 4. Discussion

The species richness of the killed vertebrates (65 species) was higher than the species richness observed in other similar studies in Amazonia [36,68]. This may be related to the high diversity of this transition area, hosting species of both Amazonia and Cerrado (data obtained from monitoring of the Sinop Hydroelectric Plant for medium and larger mammals, with 42 species; birds, with 426 species; and amphibians, with 61 species) as observed in some monitoring studies [23,34,69–71], and road features, such as location, vehicle traffic, number of lanes, and surrounding landscape [23,24,33,39].

The roadkill rate of 0.16 animals/km/day in our study was higher than that found in other studies carried out in Cerrado, such as those of Carvalho [34] (0.06 animals/km/day), Cunha et al. [23] (0.01 animals/km/day), and Melo and Santos-Filho [35] (0.13 animals/km/day), and in Amazonia, such as those by Pinheiro and Turci [37] (0.14 animals/km/day), Turci and Bernarde [36] (0.08 animals/km/day), and Junior et al. [68] (0.02 animals/km/day).



Additionally, the species richness that we found was higher, since those studies recorded between 25 and 55 road-killed species [23,34–37,68]. The sampling effort is especially decisive for the richness of road-killed species [38]. The roadkill monitoring events mentioned above were carried out with a sampling effort that ranged from 1560 to 21,600 km. However, regardless of the effort employed, the species richness and roadkill rates found in our study are superior to those found in the other studies, which can be explained by the high daily number of trucks that cross the highway.

Endangered species roadkill, such as *M. tridactyla*, *C. brachyurus*, and *P. maximus*, is commonly recorded in most studies [23,33,35,70,72–74]. The occurrence of species that are vulnerable to extinction in the region reinforces the importance of carrying out studies to evaluate fauna that has suffered road mortality and the need to create species preservation programs that guarantee permanence in their natural habitat [14].

In the wet season, the animals spread out more frequently while looking for food, breeding partners, and nesting sites [30,75], which increases mortality by roadkill, primarily for reptiles, amphibians, and birds [76–78]. However, the works by Cunha et al. [23], Melo and Santos-Filho [35], and Turci and Bernarde [36] observed a higher number of roadkills during the dry season, probably due to the characteristics of these regions such as open areas composed mainly of pasture, with few fragments of forest, occurrence of fire, scarcity of water, and increased traffic to transport agricultural products. These environmental and anthropogenic characteristics require a greater movement of animals performing their daily activities, leaving wildlife more exposed to being road-killed.

Mammals had the highest frequency of roadkill found in this work, supporting the results of many studies—e.g., [23,34,35]. The high death rate of mammals is associated with their dispersion movements, feeding, and reproductive habits [42,79,80]. For example, *C. thous* and *E. sexcinctus* were among the most abundant species in roadkill records (in this study), supporting the work of Fischer [42], Melo and Santos-Filho [35], Casella [81], Cunha et al. [23], and Carvalho [34]. These species are less sensitive to the effects of habitat fragmentation and have an omnivorous feeding habit, being able to consume carcasses of other animals [58], and are therefore attracted to the road, increasing the risk of vehicle collision.

The roadkill hotspots recorded in our study for wild mammals in built-up areas close to forest fragments and/or veredas (springs) support those reported by Casella [81], Cunha et al. [23], and Carvalho [34]. These animals can appear on sections of road with landscape features corresponding to their natural habitat [82]. Freitas [32] recorded a positive relationship between roadkill and the presence of vegetation, as well as increased road mortality in places where there were straight roads and flat topography, which tend to facilitate increased vehicle speed and therefore decrease the driver's reaction time when sighting an animal.

The frequency of bird deaths ranged from 9% to 52% in different studies [13,23,34–36,42,70,75]. Areas used for cropping or livestock serve as a major attraction for birds, as they facilitate the search for food such as insects, small animals, and grains that have fallen from trucks throughout road surface [43,64,83–85].

The bird roadkill in our study may be a result of the fact that the region has large areas of farming close to BR-163, where there are large amounts of corn and soybean deposited on the side of the road, attracting several animals, primarily birds. On the other hand, Prada [13] believes that the high mortality rate of birds occurs because the species have either diurnal or nocturnal habits, which leave them vulnerable for a longer period. The most frequent species was *Tyto furcata*, which has a nocturnal habit, representing 26% of the roadkill in our study. This species feeds mainly on rodents, which can frequent the road in search of food (grains), serving as an attraction for these birds. Rosa and Bager [78] found a positive relationship between the roadkill of birds and rice plantations, and Freitas [70] found a positive relationship with grasses and urban areas. In this study, it was noted that 96% of road sections with hotspots of bird roadkill occurred close to open areas in the

municipality of Sorriso, which is considered the largest grain producer in Brazil, supporting the results of Freitas [70] and Rosa and Bager [78].

In the case of snakes, many of the roadkills may be intentional [34], which usually occur by fear (ophidiophobia) or superstitions and other beliefs about snakes [72]. Secco et al. [86] observed drivers intentionally hitting snakes, which is common in the region (Rodrigues pers. comm.). In addition, reptiles are generally slow-moving, and many use roads for thermal regulation [18,24,87], which may have contributed to the increase in reptile roadkill records in the wet season.

Amphibians represented the group with the lowest frequency of road mortality. These values may not represent real numbers of roadkill, as monitoring carried out by means of vehicles makes their sighting difficult due to their small size [88,89], and their carcasses are easily removed by scavengers. Approximately 70% of these roadkills occurred during their reproductive period, when several species migrate to temporary pools or burrow next to the road to reproduce.

The record of endangered species roadkill and the significant hotspots observed for some animals indicate that roadkills are concentrated in some sections of the road, and these locations are highlighted for the implementation of mitigation measures. These can be faunal crossings, traffic speed reducers, and signages implemented along roadkill hotspots of BR-163 for the conservation of fauna. Therefore, understanding the dynamics of roadkill and the associated connections with landscape features and the biology of the species is an important step towards the conservation of wildlife, especially in locations with high deforestation rates and grain production, where there is a larger investment in transport infrastructure such as road duplication and railway implementation.

**Author Contributions:** Conceptualization, D.R. and E.S.; methodology, D.R.; formal analysis, M.C. and E.S.; investigation, D.R. and E.S.; resources, D.R.; data curation, D.R.; writing—original draft preparation, E.S., C.R., D.R. and M.C.; writing—review and editing, C.R., D.R. and M.C.; visualization, D.R.; supervision, D.R.; project administration, D.R.; funding acquisition, D.R. All authors have read and agreed to the published version of the manuscript.

**Funding:** This research was funded by Fundação de Amparo à Pesquisa do Estado de Mato Grosso (FAPEMAT), grant number 152004/2009", and the E.S. was funded by the Master's Program in Environmental Science and the Center for Studies on the Biodiversity of the Mato Grosso Amazonia (NEBAM).

**Institutional Review Board Statement:** The animal study protocol was approved by the Instituto Chico Mendes de Conservação da Biodiversidade (protocol code—SISBIO license number 12636, date of approval: June 2008).

**Data Availability Statement:** The data presented in this study will be openly available in a public repository after the manuscript is accepted for publication.

**Acknowledgments:** We are grateful to E. Almeida and C. Faganello for their support in the field. We would also like to give thanks to CNPq for a fellowship to D.J.R., and to Leyton Tierney and William Ramond for reviewing the English.

**Conflicts of Interest:** The authors declare no conflict of interest.

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
