# Peer review of "Hotspots and Season Related to Wildlife Roadkill in the Amazonia–Cerrado Transition"

_diversity, doi:10.3390/d14080657_

Round 1
Reviewer 1 Report
Review: Hotspots and season related to wildlife roadkill in the 2 Amazonia-Cerrado Transition.
The manuscript describes patterns of roadkill along a major highway crossing the ecotone of two very species rich biomes. The topic is of great interest for the conservation of many species as well as for the protection of this high biodiversity. The study delivers baseline data on the scope of the threat and on where and when management action is most needed. The survey methods and statistics employed are fine. I think a bit more could be done to help guide the reader through the findings and their interpretation and what steps should be taken, though what is given in the manuscript is fine. Maybe a supplementary table could be added in which the authors list the worst roadkill hotspots and their specific parametres and make suggestions on how to address those? The use of the literature and the language is fine as well.
Specific comments
L151: You state that animal carcasses were only counted in the direction of the travel. Does this mean that they were only collected from the lane in that direction, while roadkill on the lane in the opposite direction was not collected? What was done in the case of carcasses on the shoulder of the road (e.g. injured animals that managed to move away from the road itself by a couple of metres or animals that were thrown to the side after impact with the vehicle?
L234: A few carcasses were so decomposed that they could not be assigned to a vertebrate class. I assume this means they were really old. As the intervals since the previous trip would differ between trips and weather conditions etc. may influence how long such a carcase is visible, I would suggest to omit those decomposed carcasses from analysis. I actually assume this was done, but it is not mentioned in the data analysis section.
Maybe also add a percentage for the carcasses that could be identified to class but not species in the paragraph starting at L232. For birds and reptiles the percentage of such carcasses was not low.
L244-250 but also elsewhere: Consider whether to add the family or an English name for these species may help readers without specialist knowledge of the fauna of the study area to envisage the findings. At present readers would have to look up the family in Table 1.
Figures 5-8: could the main cities and the biome divide between Amazonia and Cerrado be added to these figures to help the reader envisage the patterns?
L364-377: Maybe a species accumulation (rarefaction curve) could be done to compare numbers of killed species with other studies which had surveyed different lengths of road.
Minor points:
L39: typo (bioriversity)
L40: remove ‘the’ before ‘Amazonia’
L73-74: Somehow this sentence does not fully work for me. The two parts separated by the ‘and’ seem not to fit together. Maybe the second part should read something like “and get further reduced to deaths resulting from vehicle collisions which ultimately results in loss of biodiversity.” Or did the author mean something else?
L95-96: rephrase by spelling out what “its” stands for.
Table 1: ‘chilensis’ should be lower case
L290: I would use “all classes combined” rather than just “all classes” to highlight the fact that not each single class responded in this way but the combination of all of them. The same would also apply to some other locations in this manuscript (e.g. L317). It is actually sometimes done more clearly. For example, in L301 the text refers to “all animals together” and contrasts this with the separate classes. Whatever is decided on to make the distinction clear, it should be standardized in the manuscript including the figures (e.g. the top left panel in Figure 4).
Figure 4: Top left panel ‘class’ should be plural
L475: change to “for reviewing the English”.
L534: insert space between ‘fencing’ and ‘on’
Author Response
Dear Review,
Coauthors and I are pleased to submit the revised version of manuscript "Hotspots and seasons related to wildlife roadkill in the Amazon-Cerrado Transition” for publication in Diversity Journal. We appreciate all the reviewers' suggestions, and following their suggestion, we have rewritten parts of the manuscript to provide greater clarity. We chose to send a single letter to both reviewers, as they suggested reviews that were sometimes complementary and sometimes divergent. So we think it's important that reviewers have access to the answer to both. The suggestions of the reviewers that were contradictory were answered with the one that best clarifies the work.
We will start with the review by reviewer 2, who gave more suggestions about the manuscript and encompassed part of the review by reviewer 1.
We have addressed each of their concerns as described below.
Reviewer 2 comments:
R: Done. We rewrote the sentence, adding the reviewer's suggestion.
Comment 2: Line 20 (Add the letter s in roadkill)
R: Done.
Comment 3: Line 22 (Change consists by consisted)
R: Done.
Comment 4: Line 24 (change were to was)
R: Done.
Comment 5: Line 26 (change hit to roadkill)
R: Done.
Comment 6: Line 26 (change “Endangered mammals identified” to Endangered mammals identified)
R: Done.
Comment 7: Line 29 (Add “The location and number of hotspots varied among classes and were related to different environmental variables, including...)
R: Done.
Comment 8: Lines 31 and 32 (Rephrase)
R: Done. We rewrote the sentence “Roadkills may be a major threat for vulnerable species and should therefore be studied to define conservation plans for local species and road traffic”.
Introduction
Comment 9: Lines 37 and 38. (Rephrase add Land-use change due to agr and liv exp, together with infra imple, has caused severe biodiv loss in the A-C trans zone)
R: Done.
Comment 10: Line 42. (Rephrase and order change “ of the domestic product of Mato Grosso, which is the leading state of Brazil
R: Done.
Comment 11: Line 46 (change “ with majority of destined for” to “mostly to the port of”
R: Done.
Comment 12: Line 49. Change “into” to “through” and delete from “the State….”
R: Done.
Comment 13: Lines 56-60 (Move this sentence at line 53, before "it is estimated")R: Done.
Comment 14: Lines 62-63. Delete “and ecosystems ….environments.
R: Done.
Comment 15: Lines 64-65 (Rephrase). Part of the sentence was deleted and redone.
R: Done.
Comment 16: Lines 68 and 70. We changed “die from being roadkilled” to “are killed by collisions with vehicles” and “place with high rate of” to “Roadkill hotspots”.
R: Done.
Comment 17: Lines 73 and 74. Delete from “ as the populations …. Local diversity”. However, we rewrote the sentence.
Comment 18: Line 77 (Rephrase).
We rewrote the phrase.
Comment 19: Line 83. Delete “the wet or dry”.
R: Done.
Comment 20: Line 84. Add “which show seasonal patterns related to specific phases of their life cycles”.
R: Done.
Comment 21: Line 90. Change “ effective to “measures”.
R: Done.
Comment 22: Lines 93-96 (Rephrase suggested by both reviewers).
All sentences have been changed.
R: Done.
Materials and Methods
Comment 23: Line 109. Add “road” before monitoring.
R: Done.
Comment 24: Line 112. Delete “The region ……where”.
R: Done.
Comment 25: Line 116. Delete “Data on………. on the”.
R: This was not accepted because we believe this information is important as it defines the study site.
Comment 26: Lines 118-120. Delete “and connect …….Itaúba”
R: Done.
Comment 27: Lines 121 and 123. Change “volume by number” and add being between vehicles and transport words.
Comment 28: Line 127 (Legend of figure 1).
R: This was done according to the reviewer's suggestion.
Comment 29: Lines 138-140. Delete the sentence
R: Done.
Comment 30: Lines 142, 145, 152 and 153. Delete the words marked in these lines.
R: Done.
Comment 31: (Reviewer 1). Line 151. You state that animal carcasses were only counted in the direction of the travel. Does this mean that they were only collected from the lane in that direction, while roadkill on the lane in the opposite direction was not collected?
R: Animal carcasses were recorded in both directions of the lane. The monitoring trips were carried out in alternate directions; for example, the first monitoring trip departed from Lucas do Rio Verde to Itaúba. The second departed from Itaúba to Lucas do Rio Verde, and so on.
What was done in the case of carcasses on the shoulder of the road (e.g., injured animals that managed to move away from the road itself by a couple of metres or animals that were thrown to the side after impact with the vehicle?
R: All carcasses seen both on the road and on the shoulder were recorded, and their geographical location was noted.
Comment 32: Lines 157-158.
R: We deleted the sentence as it did not make sense after the minor changes.
Comment 33: Line 166. Add “of roadkilled endangered species” after occurrence of.
R: Done.
Data analysis
All requests to rewrite sentences and delete words and/or sentences were carried out.
Comment 34: Line 224. Reviewer 2 suggested, “perhaps it would have been better to test different radii (e.g., 100, 500 and 1000 m).
R: We tested different radii (200, 500 and 100 m), and 1000 m (1 km) was the best fit due to the size of the fragments, crops, etc.
Results
Comment 35: Line 234. Reviewer 1. A few carcasses were so decomposed that they could not be assigned to a vertebrate class. I assume this means they were truly old. As the intervals since the previous trip would differ between trips and weather conditions, etc., may influence how long such a carcase is visible, I would suggest to omit those decomposed carcasses from analysis. I actually assume this was done, but it is not mentioned in the data analysis section.
R: Unidentified carcasses were not used in the statistical analyses. This information was added to the text of the manuscript.
Comment 36: Line 234. Reviewer 1. Maybe also add a percentage for the carcasses that could be identified to class but not species in the paragraph starting at L232. For birds and reptiles, the percentage of such carcasses was not low.
R: Table 1 presents the number of unidentified carcasses within each class. Many times, we found only pieces of birds and reptiles, which did not allow the identification of the family or genus.
Comment 37: Lines 239-241 and 247. Delete part of sentence and add new words.
R: In these lines, the reviewer requests the change of some words and the exclusion of others. The correction was performed as requested by the reviewer.
Comment 38: The reviewer requests a change in the legend of table 1.
R: Done
Comment 39: The reviewer requests on identification of Tyto (Tyto furcata, or Tyto alba furcata?)
R: We reviewed the specimens through photos and consultations with experts and concluded that it was Tyto furcate. This was changed in the manuscript.
Comment 40: Lines 269 a 271. Delete part of sentence.
R: Done
The comments on lines 280-281, 291-292, 294, 328, 333-335, 337-339, and 342-344 are for deleting and/or adding words. All changes have been made.
All figure legends were revised, and some of the sentences were changed for better clarity.
Comment 41 (Reviewer 1): L290: I would use “all classes combined” rather than just “all classes” to highlight the fact that not each single class responded in this way but the combination of all of them. The same would also apply to some other locations in this manuscript (e.g., L317). It is actually sometimes done more clearly. For example, in L301 the text refers to “all animals together” and contrasts this with the separate classes. Whatever is decided on to make the distinction clear, it should be standardized in the manuscript including the figures (e.g., the top left panel in Figure 4).
R: All the above suggestions were accepted and included in the manuscript.
Discussion
Comment 42: Lines 355-358: The reviewer suggest to include “this may depend on the high local diversity of this transition area, hosting species of both the Amazonia and Cerrado, and road features, such as”...
R: These lines were rewritten.
Lines 360-362 were deleted according to the reviewer’s suggestion.
In line 365, the words “of the” were changed to “carried out in the” according to the reviewer’s suggestion.
Comment 43: Lines 369-377: The reviewer suggest to include some words and to delete all sentence from line 374 – 377.
Lines 378-381 have been rewritten. The sentence on line 405 has been rewritten to meet the reviewer's request.
Comment 44: Reviewer 2: It seems that only the results of Rosa and Bager's study agrees with your findings.
R: Freitas' result also corroborates ours, although Freitas has shown greater roadkill by birds near urban areas and pastures.
Lines 432, 440, 442 and 443 were deleted sentences, and new words were added accordingly to the reviewer’s suggestion.
Comment 44: Reviewer 2: already said. It would have been interesting to see the seasonal variation in roadkill frequency at least for the main species, which would have made it possible to relate time-patterns to the life cycle and ecological requirements of each species.
R: Our goal was to focus on the mammal community. Focusing on specific species, at this point, would greatly increase the article and take the focus away from the work. We believe and defend that approaches by species are important, but it would be interesting to include other roads in this type of approach, as this makes the work more interesting.
Finally, three suggestions from reviewer 1 were not made, as in our understanding they are not relevant to the understanding of the results, at the same time they increase the size of the article unnecessarily. Are they:
- We have not added the popular names of species to the table, as we understand that for international scientific language the scientific name is sufficient. Adding one more column to the table would make it too big without adding additional relevant information;
- We tried to add the names of cities or biomes in figures 5, 6 and 7, but they were very polluted with information making it difficult to understand them, so we chose not to add this information;
- We did not perform a rarefaction figure, as we understand that this is not part of the scope of our work. In addition, the reviewer's suggestion was to add this curve to compare with other articles, but of 23 other studies analyzed, only one presented this curve. So it would not have much value for comparison with other works as suggested by the reviewer.
Thank you for the time taken to consider our revised manuscript.
Clarissa A. Rosa
Reviewer 2 Report
dear authors,
I was pleased to read your ms. I have no major concerns about both the methods and results, nonetheless I believe that the current version needs to be deeply revised to improve clarity and deliver your result more effectively.
The help of a native English speaker may help to make the style more straightforward
I suggested several changes in the attached PDF file, hoping they will be useful to improve the final draft

Author Response

(The authors gave the same response as above.)

Round 2
Reviewer 2 Report
The ms has been improved. Some minor correction are still needed, while the discussion can be improved (see suggestions in the attached file)

Author Response
Dear,
We accept all suggestions from the reviewers, including improvements to the discussion, and we have carried out yet another review of English by a native speaker. All changes are marked with the word track change.
Best
